# Antagonism of Rhizosphere *Streptomyces yangpuensis* CM253 against the Pathogenic Fungi Causing Corm Rot in Saffron (*Crocus sativus* L.)

**DOI:** 10.3390/pathogens11101195

**Published:** 2022-10-16

**Authors:** Li Tian, Shuang Hu, Xingxing Wang, Yingqiu Guo, Luyang Huang, Lili Wang, Wankui Li

**Affiliations:** Institute of Chinese Materia Medica, Shanghai University of Traditional Chinese Medicine, Shanghai 201203, China

**Keywords:** *Streptomyces yangpuensis*, corm rot in saffron, whole genome sequencing, PGPR, antagonistic activity, biological control

## Abstract

Plant diseases lead to a significant decline in the output and quality of Chinese herbal medicines. Actinomycetes play a vital role in the rhizosphere ecosystem. This is especially true for *Streptomyces*, which have become a valuable biological control resource because of their advantages in producing various secondary metabolites with novel structures and remarkable biological activities. The purpose of this study was to isolate an effective antagonistic actinomycete against the pathogen of corm rot in saffron. An antagonistic actinomycete, CM253, was screened from the rhizosphere soil samples of *Crocus sativus*, by plate co-culture with four pathogenic fungi (*Fusarium oxysporum*, *Fusarium solani*, *Penicillium citreosulfuratum*, and *Penicillium citrinum*). CM253 inhibited the growth and development of *F. oxysporum* hyphae by scanning electron microscopy (SEM) and transmission electron microscopy (TEM). Furthermore, by analyzing the degrading enzyme, the growth-promoting performance, and the whole genome of strain CM253, it was identified as *Streptomyces yangpuensis*, which produces NH_3_, protease, glucanase, cellulase, IAA, and ACC deaminase. In addition, 24 secondary metabolite synthesis gene clusters were predicted in antiSMASH. We identified genes encoding 2,3-butanediol; methionine; isoprene (*metH*, *mmuM*, *ispEFH*, *gcpE*, *idi*, and *ilvABCDEH*); biofilm formation; and colonization (*upp*, *rfbBC*, *efp*, *aftA*, *pssA*, *pilD*, *fliA*, and *dhaM*). Above all, *S. yangpuensis* CM253 showed the potential for future development as a biocontrol agent.

## 1. Introduction

Saffron is the dry stigma of *Crocus sativus* L., which belongs to the Iridaceae family. Originating from many places, such as Iran, Greece, and Spain, saffron is a world-renowned dye and spice, and is now used in the cosmetics and beverage industries [1,2,3]. Furthermore, saffron has antitumor, antioxidant, antidepressant, anti-inflammatory, and cholesterol-lowering effects [4,5]. Moreover, saffron was introduced into the mainland from India via Tibet and was gradually incorporated into the traditional Chinese medicine system. Under the system of traditional Chinese medicine, saffron also has blood- activating, stasis-dissolving, blood-cooling, detoxifying, nerve-tranquilizing, and antidepressant effects [6]. In the 1960s, saffron was introduced to China from Japan and was successfully planted in Chongming in the 1980s [7]. However, *C. sativus* is a triploid plant that can only be propagated asexually by corms, which is vulnerable to infection with viruses and is difficult to cultivate. Saffron corm rot is a global problem that has limited the development of the saffron industry [8,9,10]. Worldwide, *F. solani*, *F. oxysporum*, *F. culmorum*, *F. roseum*, *Trichoderma* sp., *Sclerotium rolfsii*, *Sclerotinia gladioli*, *Penicillium* sp., *Sclerotinia cormorum*, *Rhizoctonia crocorum*, and *Rhizopus oryzae* can cause corm rot [10,11,12,13,14,15]. In China, external factors, such as climate and soil characteristics, have led to the increased incidence of and the increased severity of saffron corm disease. This has led to a significant decrease in the saffron yield and its quality, which has restricted the development of the saffron industry [16,17,18]. For example, *F. oxysporum*, *F. solani*, *Aspergillus brasiliensis*, *Aspergillus niger*, and *Penicillium solitum* can cause corm rot in saffron [19,20,21]. In the early stage of this study, we confirmed that four strains of *F. oxysporum*, *F. solani*, *P. citreosulfuratum*, and *P. citrinum* were the main pathogens of saffron corm rot in Chongming Island—one of the main saffron-producing areas in China—among which *P. citreosulfuratum* was first reported to cause the black spot disease of saffron [22]. On Chongming Island, corm rot caused by *F. oxysporum* is the most destructive saffron disease known to date. Currently, the two-segment (TS) cropping system (indoor soilless cultivation and flower picking, and the outdoor field cultivation of seed corms) is used to reduce disease and pest impact on the *C. sativus* yield, which only partially alleviates the issue. At present, saffron, which is produced on a nationwide scale in China, cannot meet the market demand, thereby, necessitating a large number of imports from Iran.

The term “rhizosphere” was first suggested by Hilnter in 1904 and was used to define the microorganisms near the rhizosphere soil, as rhizosphere microorganisms. Rhizosphere microorganisms can be classified as beneficial, harmful, or neutral [23]. Plant growth-promoting rhizobacteria are a type of beneficial microorganism living near the plant roots, which can promote plant growth, improve crop yields and quality, and can improve plant stress resistance [24,25,26]. The healthy rhizosphere soil of plants is considered an excellent source of PGPR (plant growth-promoting rhizobacteria), which can promote plant growth and prevent diseases. Because they are affected by the secretion of medicinal plant roots over long periods of time, there are many types of rhizosphere microorganisms [27]. Actinomycetes are a type of microbial resource that can produce natural active products. Studies have shown that there are abundant actinomycetes resources in the rhizosphere of medicinal plants, which can produce a variety of antibiotics and extracellular enzymes to promote growth and protect plants from pathogens [28,29,30]. The biocontrol mechanism of *Streptomyces* on plant diseases includes antagonism, competition, and inducing plant resistance, and it is one of the main sources of known bioactive substances [31]. Various studies have shown that *Streptomyces* mediates rice blast disease, watermelon Fusarium wilt disease, Fusarium wilt of banana, and rust rot, as well as root rot in ginseng [32,33,34,35]. 

Fungicides are widely used to prevent and control corm disease in saffron and to increase the yields, which poses a risk in terms of the quality and safety of Chinese medicinal materials. Furthermore, this may also promote the emergence of drug-resistant fungal strains. Moreover, the accumulation of chemical substances also presents a risk to human health. Therefore, the use of PGPR and its metabolites, which can promote plant growth, as a biological control agent (BCA) to regulate the pathogen of corm rot in saffron may be an important tool to promote the development of the saffron industry.

The purpose of this study was to isolate an effective antagonistic actinomycete against the pathogen of corm rot in saffron. Here, 152 actinomycetes were isolated from the rhizosphere soil of healthy *C. sativus*, by co-culture with four pathogenic fungi (*F. oxysporum*, *F. solani*, *P. citreosulfuratum*, and *P. citrinum*). Next, the actinomycetes species with the best antagonistic effect was selected for further study. Using scanning electron microscopy (SEM) and whole genome sequencing analysis, *Streptomyces yangpuensis* CM253 was identified. Subsequently, the antagonistic activity and mechanism of the strain CM253 were studied and its mechanisms of biological control were evaluated.

## 2. Results

### 2.1. Screening of Biocontrol Actinomyces

In this experiment, 152 actinomycetes were isolated from the rhizosphere soil of *C. sativus*, among which the best antagonistic actinomycetes (CM253) had inhibition rates of 67.56%, 57.19%, 35.85%, and 36.36%, on *F. oxysporum*, *F. solani*, *P. citrinum*, and *P. citreosulfuratum*, respectively (Figure 1).

### 2.2. Effect of Biocontrol Actinomycete CM253 on Mycelium Morphology

In the control group, *F. oxysporum* hyphae were thick, with a smooth surface, nearly round in cross-section, clear in cell structure, and numerous in spores, all of which were indicative of a normal growth state (Figure 2A). However, in the *S. yangpuensis* CM253 treatment group, the hyphae were broken; the thickness was uneven, distorted, shriveled, and shrunken; the distribution was disordered; and the cut surface was irregular. In addition, the cytoplasm was turbid and chaotic; the cell walls, cell membranes, and dysplasia were evident in the vacuoles; and the spores were distorted and reduced (Figure 2B).

### 2.3. Analysis of Degrading Enzyme Production and Growth-Promoting Performance of Biocontrol Strain CM253

Based on the experimental phenomena, we found that strain CM253 has cellulase, protease, and glucanase activities, but lacks chitinase activity (Table 1, Appendix A). Strain CM253 can produce siderophore; IAA (indole acetic acid) and ACC (1-aminocyclopropane-1-carboxylic acid) deaminase enzyme; and NH_3_, but it cannot dissolve phosphorus and potassium (Table 1, Appendix A).

### 2.4. Identification and Whole Genome Sequencing of the CM253 Biocontrol Strain

A single colony of strain CM253 is nearly round, slightly convex, dry, with a white aerial hyphae, and a pale-yellow basal mycelium (Figure 3A). The hyphae are helical rods, and the spores are short rods, approximately 1 µm long (Figure 3B). According to the above morphological characteristics and to the physiological and biochemical tests (Table 2), this strain was preliminarily identified as *Streptomyces*. Subsequently, the 16S rRNA gene sequence was compared to the NCBI database, and strain CM253 was confirmed to be *Streptomyces*. To determine the phylogenetic status of CM253, we sequenced the complete genome of CM253 (Figure 4) (GenBank accession numbers: CP102514-CP102517). A comparison of local databases, based on 31 house-keeping genes (*dnaG*, *frr*, *infC*, *nusA*, *pgk*, *pyrG*, *rplA*, *rplB*, *rplC*, *rplD*, *rplE*, *rplF*, *rplK*, *rplL*, *rplM*, *rplN*, *rplP*, *rplS*, *rplT*, *rpmA*, *rpoB*, *rpsB*, *rpsC*, *rpsE*, *rpsI*, *rpsJ*, *rpsK*, *rpsM*, *rpsS*, *smpB*, and *tsf*), allowed for the selection of the 19 closest strains at the species level. Based on the sequence alignment and phylogenetic analysis results of 31 housekeeping genes, the similarity between the strain CM253 and *S. yangpuensis* was 98.9%, and the phylogenetic distance between strain CM253 and *S. yangpuensis* was the closest. Moreover, a phylogenetic tree was built, based on the neighbor-joining (NJ) method, using MEGA 6.0 software (Auckland, New Zealand) (Figure 5). In addition, the genome sequences of strain CM253 were 96.47% identical, by average nucleotide identity (ANI), to the *S. yangpuensis* genome. Considering the morphology, the biochemical characteristics, the phylogeneny, and the whole genome sequence analysis, strain CM253 was categorized as *S. yangpuensis*.

The complete genome of *S. yangpuensis* CM253 is composed of one chromosome (7,435,303 bp) and three plasmids (362,437 bp, 62,905 bp, and 231,470 bp); with a GC content of 72.23% (Figure 4); and a CDS value of 7237, including 73 tRNAs, 21 16S-23S-5S rRNAs, 52 sRNAs, and 7237 coding genes. The total length of the coding region is 7,053,798 bp, accounting for 87.17% of the total genome length. The average gene length is 974.69 bp, with a total of 858 TRFs, 2 pre-phages, 7 genomic islands, and 54 confirmed types of sequences predicted in the genome.

In total, 3595 coding genes of *S. yangpuensis* CM253 were annotated in the GO database, accounting for 49.68% of the total protein-coding gene sequences (Figure 6). There are 457 types of genes related to biological processes in the protein-coding genes of *S. yangpuensis* CM253, among which the largest number are the genes related to the regulation of transcription and DNA-templated (GO: 0006355,137). There are 57 types of genes related to molecular functions, of which the biggest proportion is related to membrane integration components (GO: 0016021,914). There are 798 types of genes related to cellular components, of which DNA binding (GO: 0003677, 483) is the most prevalent, followed by ATP binding (GO:0005524,352); cytoplasmic (GO:0005737,248); transcription factor activity; sequence-specific DNA binding (GO:0003700,200); metal ion binding (GO:0046872,160); and hydrolase activity (GO:0016787,154).

The protein coding sequences of the predicted genome of *S. yangpuensis* CM253 was compared with the COG database, and we found that 5380 protein-coding genes could be functionally annotated, accounting for 74.34% of the total protein sequence. There are 500 metabolic pathways involved in transcription, 372 pathways involved in amino acid transport and metabolism, 291 pathways involved in energy production and conversion, 289 pathways that participate in signal transduction mechanisms, and 284 pathways that participate in hydrocarbon transport and metallurgy. Only a small number of the COG pathways have clear functions, while the other 1989 pathways with unknown functions need to be further studied in the future (Figure 7).

A total of 3128 genes were annotated in the *S. yangpuensis* CM253 genome, in the KEGG database, accounting for 43.22% of the predicted number of genes, and 42 functional entries, with the gene functions divided into six classes. Genes related to metabolism were the most prevalent—up to 2343—including global and overview maps (902); amino acid metabolism (288); carbohydrate metabolism (271); metabolism of cofactors and vitamins (171); and energy metabolism (165); as well as other pathways, which is conducive to growth in complex environments and broad-spectrum antibacterial ability. The number of genes annotated to environmental information processing, human diseases, genetic information processing, cell processes, and organizational systems were 225, 107, 219, 150, and 84, respectively (Figure 8).

Comparing the genome sequence with the CAZy database, we found that there were 76 protein-coding domains in the *S. yangpuensis* CM253 genome belonging to the CAZy family. These include 76, 60, 59, 28, 8, and 1 gene from the following groups: glycoside hydrolases (GHs), glycolipids (CES), glycosyl transferases (GTs), auxiliary activities (AAs), carbohydrate-binding modules (CBMs), and polysaccharide lyases (PLs), respectively. The most abundant family is GT2 (24 members), which is typically responsible for transferring nucleotide diphosphate sugars to polysaccharides and lipids (Figure 9).

There are 24 secondary metabolite biosynthesis gene clusters in the *S. yangpuensis* genome, including terpene, non-ribosomal peptide-synthase (NRPS), polyketide synthase (PKS), siderophore, lanthipeptide, melanin, and bacteriocin. Four of these are identical to the synthetic gene clusters of venezuelin, JBIR-126, coelichelin, and alkylresorcinol with antibacterial activity. Among these, five predicted gene clusters have no similarity with known gene clusters, and these gene clusters may also synthesize some new active substances; however, all of these gene clusters require further verification (Table 3).

## 3. Discussion

In this study, the CM253 strain was isolated from the rhizosphere soil of healthy *crocus sativus*, which had significant antagonistic effects on four *C. sativus* pathogens. We previously isolated *Pseudomonas aeruginosa*, which also had an obvious antagonistic effect on the pathogenic fungi of saffron [36]. There have been numerous studies on the prevention and control of crocus corm rot and the advancement of crocus yields and quality by beneficial strains. For example, *Burkholderia gladioli* E39CS3, an endophytic bacteria of *C. sativus*, has been shown to inhibit the in vivo corm rot of *F. oxysporum*, a specific pathogen of *C. sativus*, and to induce systemic drug resistance (ISR) [37]. *Curtobacterium herbarum*, Cs10, improves the number of flowers and significantly enhances the length of the saffron filaments and the overall saffron production [38]. Based on morphology, biochemical characteristics, phylogeny, and a whole genome sequence analysis, the CM253 strain was identified as *S. yangpuensis*. Importantly, *S. yangpuensis* is reported to have an inhibitory effect on *F. oxysporum*. Furthermore, to the best of our knowledge, this is the first study demonstrating that *S. yangpuensis* has an antagonistic effect on *P. citreosulfuratum*, *F. solani*, and *P. citrinum*. *S. yangpuensis* also has strong antagonistic activity against *Botryosphaeria dothidea*, *Fusarium pseudograminearum*, *Phytophthora capsici*, *Colletotrichum orbicule*, *Gaeumannomyces graminis var. tritici*, and *Setosphaeria turcica* [39].

The phenomena of plant nitrogen deficiency, phosphorus deficiency, and iron deficiency occur widely in nature and are three important factors that limit plant growth. An evaluation of its PGP performance showed that CM253 carries out and produces iron carriers and NH_3_. Siderophore is an organic chelating agent with a relatively low molecular weight (500–1000), which can combine with insoluble iron elements in the environment and form siderophore-Fe to prevent the proliferation of plant pathogens and to effectively promote antibiotic entry [40]. A bioinformatic analysis of the CM253 genome, using antiSMASH, showed that three biosynthesis gene clusters of iron were composed of 26 genes. Among them, *iucD* belongs to the NIS-type iron carrier biosynthetic gene, while *hemE* and *hemY* exist in the porphyrin metabolic pathway, which have the functions of protoporphyrin/coproporphyrin ferrochelatase and Fe-coproporphyrin III decarboxylase, respectively. The amino acid sequences in *hemE* and *hemY* gene are 384 bp and 494 bp in length, respectively. ACC deaminase catalyzes ethylene in plants to α-ketobutyrate, thus reducing the ethylene-mediated inhibition of photosynthesis. The gene, gene4579, which regulates the metabolic pathway of ACC degradation to 2-Oxobutanoate, was discovered in the CM253 genome. IAA is a plant hormone that regulates plant root growth by stimulating the proliferation and elongation of root cells. The IAA synthesis pathway is divided into the tryptophan-dependent pathway and the tryptophan-independent pathway. The *trpABC*, *aofH*, and *amiE* involved in regulating the tryptophan-dependent IAA synthesis pathway were found in the CM253 genome [41,42]. The PGP performance evaluation test data also directly showed that *S. yangpuensis* can produce IAA and ACC deaminase, which has the potential to promote the root development of *C. sativus.*

The main mechanisms of bacterial antagonism include competition for nutrients and space, antibiotic synthesis, and the induction of host resistance [43]. The CM253 genome contains genes related to protease, cellulase, and β-glucosidase, such as *pepD*, *clpPX*, *dop*, *tri*, *ftsH*, *lon*, *bglBX*, and *xynA*, as well as some unknown genes, which can damage the cell walls of pathogenic fungi [44,45,46]. H_2_S can help plants resist various abiotic stresses such as drought, cold, heat, salinity, hypoxia, and toxic metals [47]. The *cysCDHKMNZ* gene, which is related to sulfate transport, was also identified in the CM253 genome [48]. Biofilm formation is the first step in root colonization by rhizosphere bacteria, which can help to mediate plant protection from environmental factors and plant diseases. Therefore, the discovery of genes related to the biofilm formation and colonization of CM253 is important for aiding our understanding of its mechanism as a biocontrol bacterium [49]. In this study, we found related biofilm formation and colonization genes in the CM253 genome, including *upp*, *rfbBC*, *efp*, *aftA*, *pssA*, *pilD*, *fliA*, and *dhaM* [50]. The synthetic genes (*metH*, *mmuM*, *ispEFH*, *gcpE*, *idi*, and *ilvABCDEH*) of 2,3- butanediol, methionine, and isoprene were also found in the *S. yangpuensis* genome, which suggests that *S. yangpuensis* CM253 may induce host resistance by producing resistance-related substances to fight the stem rot of *C. sativus*. The prediction data of the gene clusters synthesized by secondary metabolites, suggest that the similarity of four gene clusters with the synthesis of venezuelin, alkylresorcinol, JBIR-126, and coelichelin gene clusters with antibacterial activity, was 100% [51,52,53]. These antibacterial substances may be related to the inhibitory effect of CM253 on saffron pathogens. Therefore, there is a need to isolate, purify, and identify the antifungal metabolites produced by this strain to clarify its antagonistic mechanism against the pathogens of corm rot in saffron crocus pathogenic fungi.

## 4. Materials and Methods

### 4.1. Indicative Pathogenic Fungi

In 2020, four pathogenic fungi (*F. oxysporum*, *F. Solani*, *P. citreosulfuratum*, and *P. citrinum*) were isolated and purified from the rotten bulbs of saffron, according to Koch postulates, and were stored in our laboratory’s 4 °C refrigerator.

### 4.2. Collection of Soil and Isolation and Purification of Biocontrol Actinomycetes

The rhizosphere soil samples of *C. sativus* were collected from Chongming District, Shanghai (31.62° N and 121.40° E), in December 2020. They were stored in plastic bags, in refrigerated containers, and transported to the laboratory of the Institute of Traditional Chinese Medicine, Shanghai University of Traditional Chinese Medicine, for actinomycetes isolation. After the rhizosphere soil was naturally dried in the shade, it was crushed in a mortar, screened using a 20-mesh sieve, and 5 g of soil samples were weighed and placed in a triangular bottle, containing 45 mL of sterile water. Next, a sterile triangular bottle was placed on a shaker at 30 °C, sealed with a sealing film, and shaken for 2 h at 150 rpm [54]. Subsequently, 1 mL was taken and diluted in 9 mL of sterile water, in a centrifuge tube, and thoroughly mixed. This mixture was used to prepare 10^−2^, 10^−3^, 10^−4^, and 10^−5^ soil suspensions for later use [55]. Approximately 200 μL of each serially diluted suspension was added to Gause′s medium No. 1, trehalose–proline medium, HV medium, R2A medium, ISP2 medium, ISP3 medium, and ISP5 medium, containing cycloheximide (50 mg/L) and nalidixic acid (20 mg/L), respectively, which were used to cultivate actinomycetes [56,57,58]. This was repeated 3 times and all the plates were stored upside down and colony growth was observed every day. Actinomycetes colonies were picked and transferred to ISP3 culture medium. After purification, single colonies were inoculated into the corresponding slant culture medium and stored at 4 °C, for later use.

### 4.3. In Vitro Screening for Antibacterial Activity

The isolated actinomycetes were screened in vitro by the plate confrontation method [59]. Single colonies of 5 mm *F. oxysporum*, *F. solani*, *P. limonene sulfate*, *P. citrinum*, and actinomycetes were punched with a hole punch, then the pathogen plugs were placed in the center of PDA plate. Subsequently, actinomycetes were inoculated approximately 2.5 cm away from the pathogen plugs in a criss-cross formation and compared with the PDA plate with only pathogen plugs. The plates were then cultured in an incubator at 28 °C, for 5–10 days, before they were evaluated for the appearance of antagonistic bands in the experimental group. Each bacterium was tested in technical and biological triplicates and the inhibition rate was calculated to identify the actinomycetes with the best antagonistic activity. The zone of inhibition against pathogenic fungi was estimated by averaging two diameters, measured perpendicularly. Inhibition rate = (control colony diameter − treatment colony diameter)/control colony diameter × 100% [60].

### 4.4. Morphological Characteristics, Physiology, and Biochemistry of Rhizosphere Actinomycete Strain CM253

Actinomycetes were streaked onto the ISP3 medium for 5 days, and the shape, color, and texture of colonies were recorded. Square agar blocks with actinomycetes, with a side length of approximately 0.5 cm, were cut from the medium, and SEM samples were prepared according to a previously described method [61]. Briefly, the mycelium was fixed with 2.5% (*v*/*v*) glutaraldehyde, at 4 °C, for 24 h, washed three times with 0.1 M PBS, then fixed with 1% (*v*/*v*) osmic acid at 4 °C, for 3 h, and washed three times with 0.1 M PBS. Next, samples were centrifuged at 8000 rpm for 10 min in 30%, 50%, 75%, 90%, 95%, and 100% (*v*/*v*) absolute ethanol, for gradient dehydration. Samples were then dried in a Leica EM CPD300 critical point dryer (Leica Microsystems GmbH, Wetzlar, Germany), and the dried sample was attached to a special sample table, with double-sided conductive adhesive. Finally, the sample table was placed in a Leica EM ACE600 ion sputter coater (Leica Microsystems, Vienna, Austria) to finish the surface gold plating treatment, and further observed under SEM. Physiological and biochemical tests of biocontrol actinomycetes were carried out according to the handbook of identification of common bacterial systems [62]. Production of H_2_S and measurement of growth temperature, starch hydrolysis, gelatin liquefaction, and nitrate reduction were assessed, among other characteristics.

### 4.5. Molecular Identification of Rhizosphere Actinomycete Strain CM253

PCR amplification of the 16S rRNA gene in CM253 was achieved using universal primers 27 F: 5′-AGAGTTTGATCCTGGCTCAG-3′ and 1492R: 5′-TACCTTGTTACGACTT-3′ [63]. The amplified PCR products were then purified and sequenced by Sanger sequencing at Biotechnology Co., Ltd. (Shanghai, China). The 16S rRNA sequence was compared to the NCBI GenBank (http://www.ncbi.nlm.nih.gov/, accessed on 22 August 2022) database, using the blast (https://www.ncbi.nlm.nih.gov/BLAST, accessed on 22 August 2022) [64] algorithm. A phylogenetic tree was then constructed using the NJ method in the MEGA 6.0 software (Auckland, New Zealand). Support for the tree topology was estimated with 1000 bootstrap replicates [65].

### 4.6. Effect of Rhizosphere Actinomycete Strain CM253 on Mycelia Morphology of F. oxysporum

*F. oxysporum* was inoculated on PDA culture medium and grown in an incubator at 28 °C, for 5–10 days. Microscopic characteristics were observed by SEM, according to the above method. Thereafter, 1 mm^3^ hyphae samples were taken and used to prepare TEM samples, as previously described [66]. Briefly, the mycelium was fixed with 2.5% (*v*/*v*) glutaraldehyde, at 4 °C, washed with 0.1 M PBS (pH 7.2) three times, fixed in 1% (*w*/*v*) osmic acid for 2 h, and rinsed with 0.1 M PBS (pH 7.2) three times. Subsequently, samples were embedded in epoxy resin. Finally, sections were cut into 60 nm slices, using a Leica 705902 ultrathin microtome (Leica Microsystems, Wetzlar, Germany), before staining with lead citrate for 15 min. Finally, samples were rinsed three times with ddH_2_O and left to air dry before analysis on a Feitecna AI G2 Spirit TEM.

### 4.7. Determination of Hydrolase Activity

Secretion of the lytic enzymes is considered to be an effective way to prevent plant pathogens near the rhizosphere, among all known biological control mechanisms [67]. A single colony of the activated biocontrol bacteria was selected and connected to the chitin assay plate, prepared as previously described by Lau et al. [68]. Glucanase-, protease-, and cellulase assay plates were prepared as previously described by Putri et al. [69], Prajapati et al. [70], and Sukmawati et al. [71], respectively. A transparent circle was indicative of the bacteria being able to produce the relevant degrading enzyme.

### 4.8. In Vitro Assessment of Plant Growth Promotion (PGP) Traits

Using the method described by Suárez-Moreno et al. [72], phosphate solubilizing capacity was qualitatively determined by inoculating a single colony of each strain in the phosphate growth medium (NBRIP) from the American Botanical Institute, for 5 days. According to the method described by Chengqun et al. [73], the biocontrol bacteria were inoculated on a silicate medium plate for five days, and a clear circle was produced, suggesting that the bacteria can dissolve potassium. Referring to the method described by Suarez-Moreno et al. [72], the specific protocol was as follows: biocontrol actinomyces were inoculated into LB liquid medium, containing 100 mg/L-tryptophan. After culturing for 3 days, 100 μL of liquid bacterial culture was transferred to a test tube and mixed with an equal volume of Salkowski colorimetric solution (30 min of 35% (*v*/*v*) HClO_4_ + 1 mL of 0.5 mol/L FeCl_3_), before incubation in the dark for 30 min. If the color turned red, this suggested that the actinomyces could produce IAA. As described by Dubey [74], the biocontrol actinomycetes were inoculated in nitrogen-free liquid medium for 1 day, and 100 μL of bacterial liquid was inoculated into DF medium for shaking culture, for 2 days, and then transferred to ADF medium, according to 2% of the inoculum. Cultivation continued for 2 d. Normal growth was indicative of ACC (1-aminocyclopropane-1-carboxylic acid) deaminase production. The biocontrol actinomycetes were inoculated in the CAS medium for five days, and an orange halo was produced, suggesting siderophore production [75]. The biocontrol actinomycetes were inoculated into 50 mL peptone water (10 g/L) in conical flasks, and after 3 d of culture, 2 mL of Nessler reagent was added to determine ammonia production in peptone water [76].

### 4.9. Whole Genome Sequencing (WGS) of Biocontrol Actinomycete CM253

#### 4.9.1. Sample Preparation

A single colony of biocontrol actinomycete CM253 was selected and inoculated into GY liquid medium, which was then cultured at 30 °C, with shaking at 200 rpm. Once the cultures reached an optical density of 0.4–0.8 at 600 nm (OD_600_), the cells were harvested by centrifugation at 8000 rpm, at 4 °C, for 5 min. Next, an appropriate amount of PBS buffer solution was added, and the cells were centrifuged again before being quickly frozen in liquid nitrogen. The cells were immediately transported by dry ice to MGE Biological Medicine Technology Co., Ltd (Shanghai, China), for whole genome sequencing.

#### 4.9.2. DNA Extraction, Genome Sequencing, and Assembly

Genomic DNA was extracted using the Cetyltrimethyl Ammonium Bromide (CTAB) method, with minor modifications. Following DNA concentration, the quality and integrity were determined by a Qubit Flurometer (Invitrogen, Carlsbad, CA, USA) and a NanoDrop Spectrophotometer (Thermo Fisher Scientific, Waltham, MA, USA).

For Illumina sequencing, at least 1 μg of genomic DNA was used for each strain for sequencing library construction. DNA samples were sheared into 400–500 bp fragments, using a Covaris M220 Focused Acoustic Shearer. Fragment size distribution was determined using agarose gel electrophoresis and genomic fragments in the 300–500 bp range were enriched for Illumina sequencing libraries, using the NEXTflex™ Rapid DNA-Seq Kit (Bioo Scientific, Austin, Austria) [44]. At least 15 μg of genomic DNA was processed into 10 kb fragments, using G-tubes (Covaris, Inc., Woburn, MA, USA). These fragments were then purified according to instructions (Pacific Biosciences, Menlo Park, CA, USA), with terminal completion and two ends connected to the SMRT bell sequencing linker, respectively [77]. Genomic DNA was sequenced using a combination of PacBio RS II Single Molecule Real Time (SMRT) and Illumina sequencing platforms. All analyses were performed using the free Majorbio Cloud Platform (www.majorbio.com accessed on 6 December 2021), from Shanghai Majorbio Bio-pharm Technology Co., Ltd.

The complete genome sequence was assembled using both PacBio and Illumina reads. The original image data were transferred into sequence data via base calling, which is defined as raw data or raw reads, and saved as FASTQ files. These FASTQ files were the original data provided for users, in which the read sequences and quality information were included. A statistic of quality information was applied for quality trimming, by which the low-quality data could be removed to generate clean data. The reads were then assembled into contigs, using hierarchical genome assembly process (HGAP) and canu [78]. The last circular step was checked and finished manually, generating a complete genome, with seamless chromosomes and plasmids. Finally, error correction of the PacBio assembly results was performed using the Illumina reads, using Pilon [79].

#### 4.9.3. Gene Prediction and Annotation

Glimmer [80] was used to predict the coding sequence in the genome, and tRNAscan-SE, RNAmmer, Rfam, and CRISPR recognition tools were used to predict tRNA, rRNA, ncRNA, and CRISPRs, respectively [81,82,83]. Next, functional annotation was performed by comparison with databases of the Cluster of Orthodox Groups (COG), Kyoto Encyclopedia of Genes and Genomes (KEGG), and Gene Ontology (GO). Secondary metabolites and carbohydrate-active enzymes were predicted using antiSMASH 4.0.2 (https://dl.secondarymetabolites.org/releases/4.0.2/, accessed on 6 December 2021) and CAZymes databases [84]. Finally, the genome circle map was drawn in CGview [85].

## 5. Conclusions

We selected an effective actinomycete, CM253, from the rhizosphere soil of *crocus sativus*. According to the above experimental results, we found that *S. yangpuensis* produces NH_3_, protease, glucanase, cellulase, IAA, and ACC deaminase to achieve biological control of the pathogenic fungi of *C. sativus* corm rot. According to the analysis of the synthetic gene cluster of the secondary metabolites, we found that CM253 had a very active metabolic capacity. There is a need to isolate and purify the antifungal metabolites produced by CM253 in future studies, to clarify its antagonistic mechanism against the pathogen of corm rot in saffron crocus pathogenic fungi.

## Figures and Tables

**Figure 1 pathogens-11-01195-f001:**
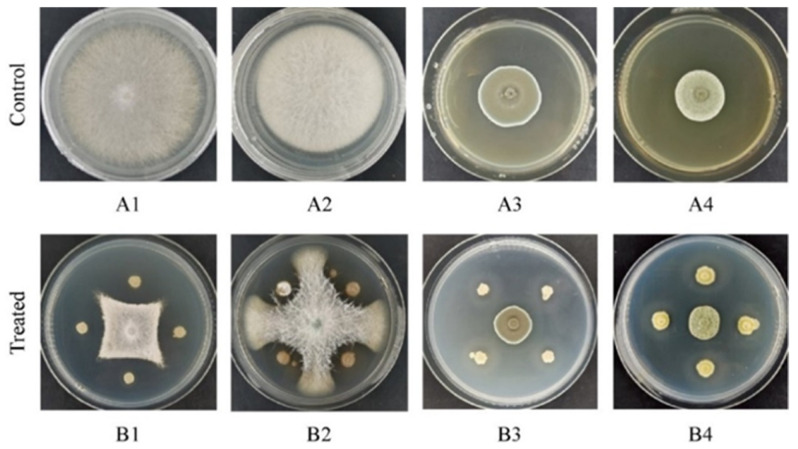
Antagonism of the strain on four pathogenic bacteria in PDA culture medium for approximately 10 days: (**1**) *F. oxysporum*, (**2**) *F. solani*, (**3**) *P. citrinum*, *and* (**4**) *P. citreosulfuratum.* (**A**) control and (**B**) treated.

**Figure 2 pathogens-11-01195-f002:**
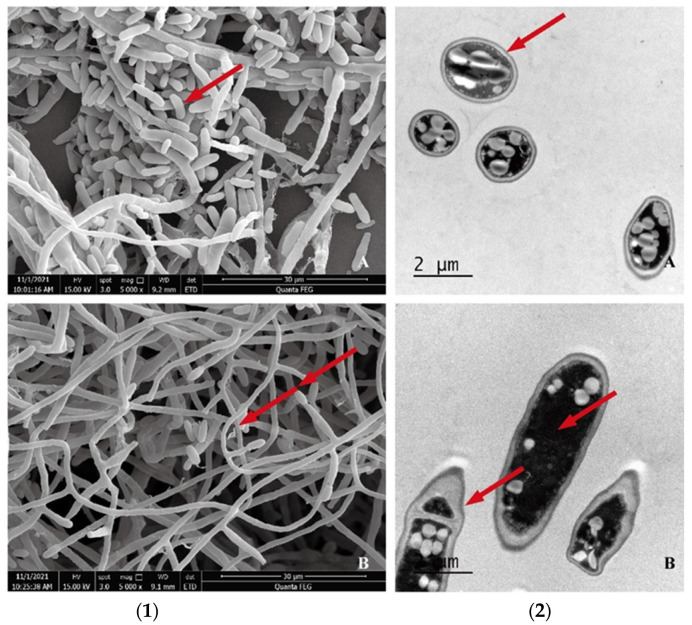
(**1**) SEM observation of *F. oxysporum*. (**2**) TEM observation of *F. oxysporum*: (**A**) contrast and (**B**) experimental. Hyphae were thick, with a smooth surface, nearly round in cross-section, and clear in cell structure (arrowhead in **1A**). Hyphae were distorted, shriveled, and shrunken (arrowheads in **1B**). The cytoplasm was turbid and chaotic; the cell walls, cell membranes, and dysplasia were evident in vacuoles (arrowhead in **2B**), compared to the contrast (arrowhead **2A**).

**Figure 3 pathogens-11-01195-f003:**
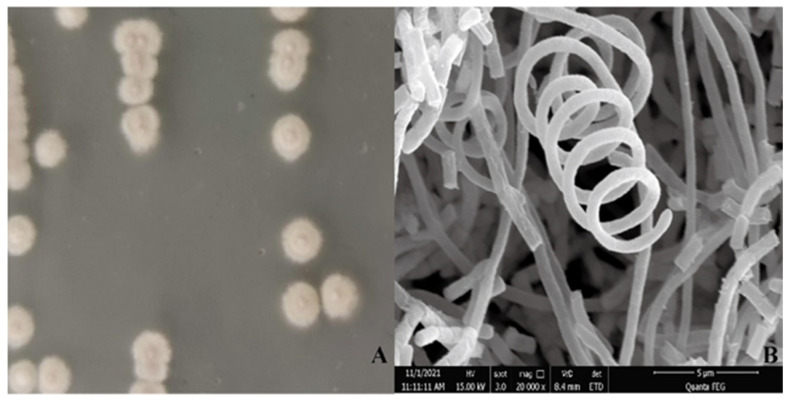
Morphology of *S. yangpuensis* CM253. (**A**) Colony morphology on ISP3 medium: the single colony is nearly round, slightly convex, dry, with a white aerial hypha, and a pale-yellow basal mycelium. (**B**) Microscopic morphology: the hyphae are helical rods.

**Figure 4 pathogens-11-01195-f004:**
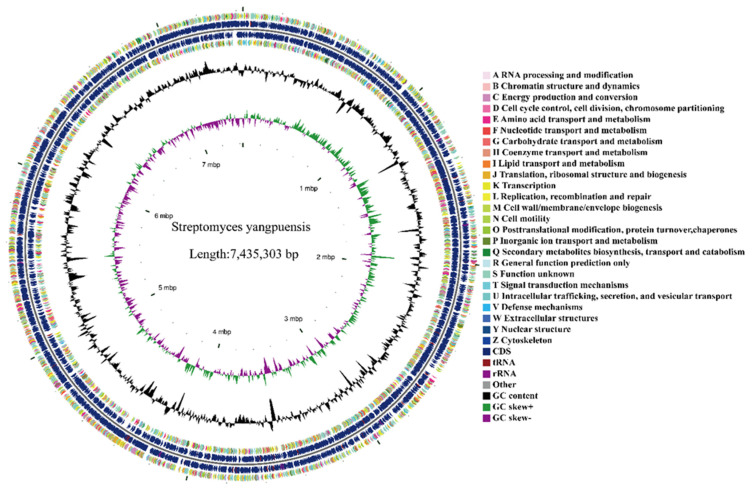
*S. yangpuensis* CM253 genome circle diagram: From the outside to the inside, the first circle and the fourth circle are CDS on the positive chain and negative chain, and different colors indicate different COG functional classifications. The second circle and the third circle are CDS, tRNA, and rRNA, on the positive chain and negative chain, respectively. The fifth circle is GC content, and the sixth circle is GC-skew value.

**Figure 5 pathogens-11-01195-f005:**
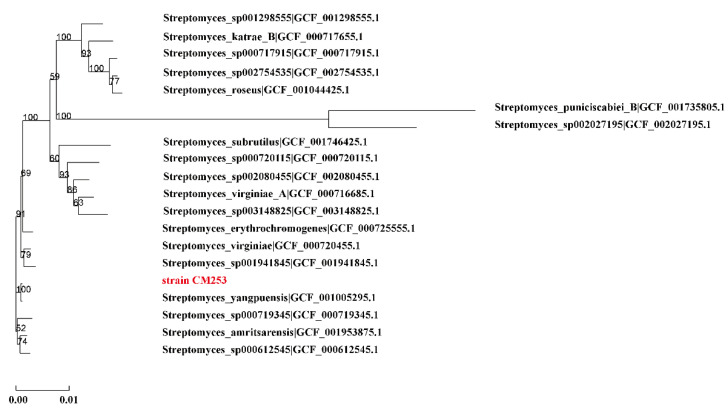
Phylogenetic tree of house-keeping genes of strain CM253. A phylogenetic tree was built, based on the neighbor-joining (NJ) method, using MEGA 6.0 software, based on 31 house-keeping genes (*dnaG*, *frr*, *infC*, *nusA*, *pgk*, *pyrG*, *rplA*, *rplB*, *rplC*, *rplD*, *rplE*, *rplF*, *rplK*, *rplL*, *rplM*, *rplN*, *rplP*, *rplS*, *rplT*, *rpmA*, *rpoB*, *rpsB*, *rpsC*, *rpsE*, *rpsI*, *rpsJ*, *rpsK*, *rpsM*, *rpsS*, *smpB*, and *tsf*). The numbers at the nodes indicate the levels of bootstrap support (%), based on 1000 reassembled datasets.

**Figure 6 pathogens-11-01195-f006:**
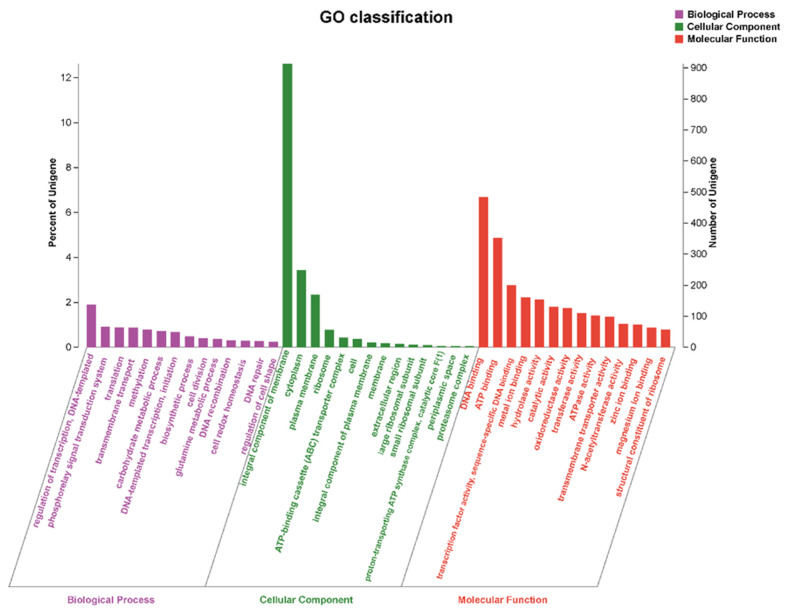
GO function classification of *S. yangpuensis* CM253.

**Figure 7 pathogens-11-01195-f007:**
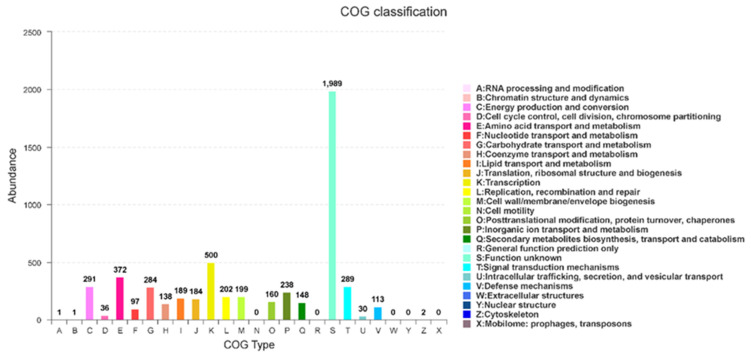
COG function classification of *S. yangpuensis* CM253.

**Figure 8 pathogens-11-01195-f008:**
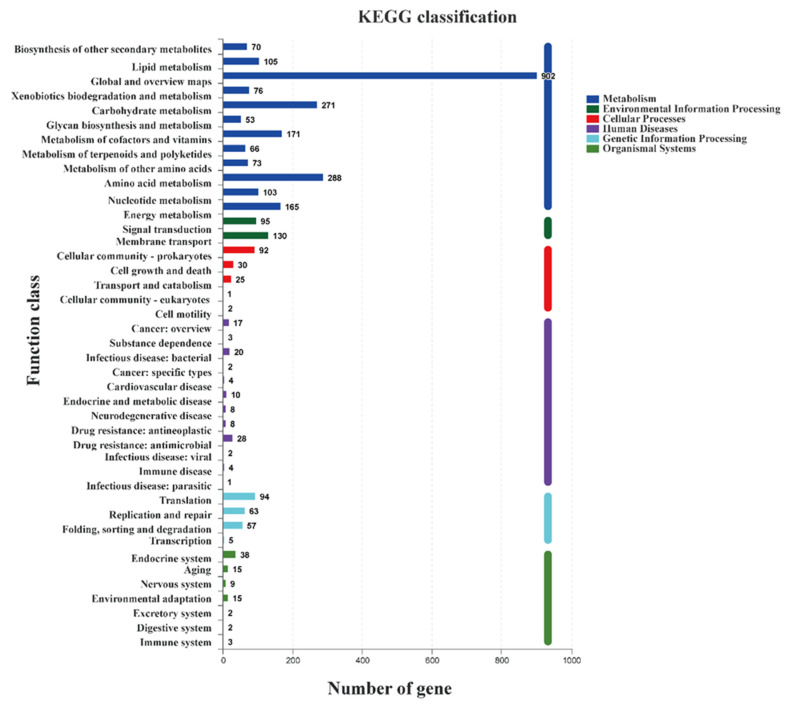
KEGG function classification of *S. yangpuensis* CM253.

**Figure 9 pathogens-11-01195-f009:**
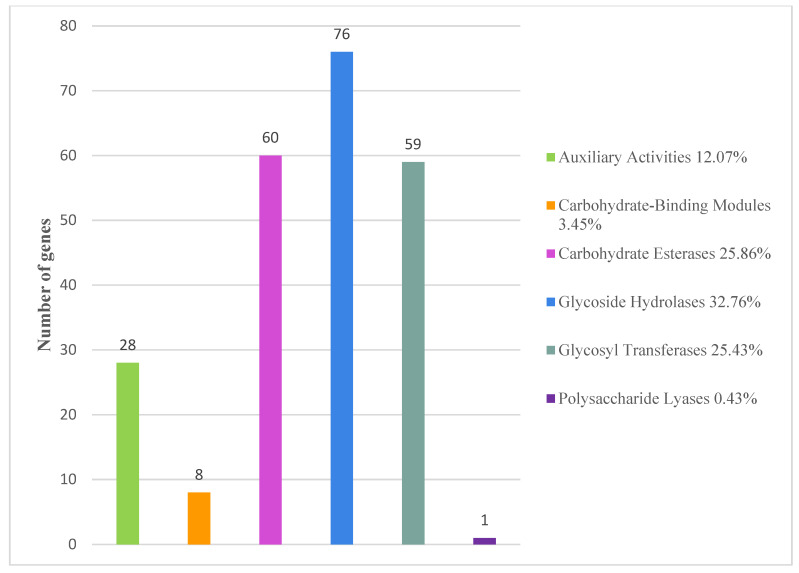
Gene count distributions of carbohydrate-active enzyme (CAZy) families.

**Table 1 pathogens-11-01195-t001:** Hydrolytic enzyme and PGP activities of *S. yangpuensis* CM253.

Hydrolytic Enzyme and PGP Activities	Result
Chitinase	−
Cellulase	+
Protease	+
Glucanase	+
IAA production	+
Phosphate solubilization	−
NH_3_ production	+
ACC deaminase enzyme	+
Potassium dissolution	−
Siderophore production	+

Abbreviations: +—positive for test; −—negative for test.

**Table 2 pathogens-11-01195-t002:** Physiological and biochemical characteristics of *S. yangpuensis* CM253.

Physiological and Biochemical Characterization	Result
Gram reaction	+
Growth at 4 °C	−
Growth at 4 °C	−
Oxidase	−
Catalase	−
Urease	+
Lipase	+
Voges–Proskauer	−
Methyl red	−
Starch hydrolysis	−
Gelatin liquefaction	−
Nitrate reduction	+
Citrate utilization	−
H_2_S production	+
Milk coagulation	−

Abbreviations: +—positive for test; −—negative for test.

**Table 3 pathogens-11-01195-t003:** Results of the antimicrobial gene clusters in the CM253 genome.

Region	Type	Start	End	Similar Cluster	Similarity
1	Lanthipeptide	23898	45961	Venezuelin	100%
2	Lanthipeptide	52651	77063	-	–
3	Terpene	427823	452809	Carotenoid	63%
4	NRPS	475101	526320	Deimino-antipain	66%
5	T2PKS	529223	601766	Spore pigment	66%
6	NRPS	897035	964537	JBIR-126	100%
7	Siderophore	2716464	2727579	Desferrioxamin B/Desferrioxamine E	83%
8	NRPS	4016003	4057260	Phosphonoglycans	3%
9	Siderophore	5560082	5573594	-	-
10	Bacteriocin	5869688	5881185	-	-
11	Terpene	5964827	5986769	Toxoflavin/fervenulin	14%
12	T1PKS	6014531	6059440	Herboxidiene	9%
13	Terpene	6308285	6335005	Hopene	61%
14	NRPS-like	6526608	6569022	Polyketomycin	4%
15	NRPS	6704968	6745944	Streptothricin	87%
16	Terpene	6838051	6857502	–	–
17	NRPS	6860761	6911849	Coelichelin	100%
18	Terpene	6937552	6957566	Ebelactone	5%
19	Terpene	7007207	7028140	Monensin	5%
20	Melanin	7031380	7057085	Istamycin	4%
21	T1PKS	7057399	7102148	Daptomycin	4%
22	Siderophore	7138154	7151437	–	–
23	T3PKS	7184657	7225715	Alkylresorcinol	100%
24	CDPS	7237760	7287223	Cosmomycin C	5%

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
