# Peer review of "Antagonism of Rhizosphere *Streptomyces yangpuensis* CM253 against the Pathogenic Fungi Causing Corm Rot in Saffron (*Crocus sativus* L.)"

_pathogens, 2022, doi:10.3390/pathogens11101195_

Round 1

Reviewer 1 Report

Please find my comments in the attached pdf file.

Good luck

Reviewer 2 Report

There is increasing demand toward effective non-chemical plant protection methods. The manuscript reports about a new Streptomyces yangpuensis strain with promising biocontrol potential against corm rot of saffron. a recent publication from (almost) the same authors was published last year in the Pathogens: In Vitro Study of Biocontrol potential of rhizospheric Pseudomonas aeruginosa against pathogenic fungi of saffron (Crocus sativus L.)  (doi: 10.3390/pathogens10111423). This manuscript has similar structure and methodology to provide information about another rhizospheric bacteria.

Minor remarks:

·         Line 27: NH3 – should be corrected to NH3

·         Line 49: Publication detailes (e.g. online availability) of the mentioned “Board of Pharmacopeia of the People’s Republic of China, 2020 should be provided.

·         Lines 58-59: only fungal pathogens are listed, and one of the cited literature, Mizusawa et al 1923, No.15 is describibg only bacterial pathogens.

·         Lines 63-64: Anthracnose is a disease, not a genus name. Please correct.

·         Lines 99-100: The sentence should be clarify the way of isolation of actinomycetes..

·         Line 117: Please add the strain No of Streptomyces yangpuensis

·         Figure 2: What do the arrows indicate?

·         Lines 123-124: Please provide the species name of the microbe.

·         Line 132: Please indicate the species name also, not only the number of the strain.

·         Table 1: Whole name of IAA and ACC should be provided.

·         Figure 3: Please indicate the species name also, not only the number of the strain. Provide details about cultivation characteristics for picture A, provide details for B (e.g. SEM).

·         Figure 4:Please indicate your strain.

·         Figure 4 legend: Please provide details about tree construction (e.g. genes used for the analysis, way of tree contraction, considering gaps etc.).

·         Figure 3: Please indicate the species name also, not only the number of the strain. 3 and – are not abbreviations. V-P and MR however abbreviations, whote names shoud be indicated.

·         Figures 5 and 6: Please indicate the species name also, not only the number of the strain in the legend of the figures. Magnification of letters are recommended.

·         Figure 9: Title in the figure should be cut off. More information is necessary in the legend.

·         Lane 228: Cited literature indicate pathogenic fungi, not bacteria

·         Lane 321: Burkholderia is bacrerium, nut fungus

·         Lane 240: Neither different strains (AIS-2, 8, 10), nor plat growth test was described previously in the results. Please provide further data, or delete them.

·         Lanes 269-270: Please provide reference.

·         Lanes 272-273: Antibiotics are secondary metabolites, it is not necessary to mention both.

·         Lane 290: please order 53-55 citation based on their publication date.

·         Lanes 296-297: Please provide further details about tested pathogens about their origin and accurate identification.

·         Lane 305: Was the triangular bottle previously sterilized? Was it covered during shaking?

·         Lanes 308-311: Please provide reference.

·         Lane 321: Sulphate is necessary to be mentioned?

·         Lane 325: I think fungal should be written instead of “bacterial”.

·         Lanes 238-331: Please provide detailes, how could you determine diameter following crisscross inoculation?

·         Berger handbook of bacterial identification cannot be found in reference No 61. Please provide appropriate reference for identification based on physiological and biochemical tests.

·         Lanes 350-351: Please provide more details fot the phylogenetic tree construction, including software, method, etc.

Round 2

Reviewer 2 Report

The authors answered all my questions, and carefully changed the manuscript according to comments.

Some minor remarks:

Lane 145: A. niger is mentioned first time here at the manuscript, please write the whole genus name.

Lane 147: The following pathogens has been mentioned previously: Fusarium oxysporum, Fusarium solani, Penicillium citreosulfuratum, and Penicillium citrinum Please use their short name here: F. oxysporum, F. solani, P. citreosulfuratum, and P. citrinum  

Lane 149: see my previous remark and use P. citreosulfuratum and not the whole name

Lane 151: see my previous remark and use F. oxysporum and not the whole name

Lane 163:  PGPR (Plant growth-promoting rhizobacteria) capital is not necessary in „Plant”

Lane 185: see my previous remark and use F. oxysporum, F. solani, P. citreosulfuratum, and P. citrinum, and not the whole name

Lane 241: Please correct to „(A) Conntrast” (capital, like in „(B) Experimental”)

Lanes 241-244: Missing information of arrowheads in pictures 1A, 2A, 1B.  Suggestion for correction: Figure 2. (1) SEM observation of F. oxysporum. (2) TEM observation of F. oxysporum, (A) Contrast, (B) Experimental. Hyphae were thick, with a smooth surface, nearly round in cross-section, and clear in cell structure (arrowhead in 1A). Hyphae were distorted, shriveled, and shrunk (arrowheads in 1B). The cytoplasm was turbid and chaotic; the cell walls, cell membranes, and dysplasia were evident in vacuoles (arrowhead in 2B) compared to the contrast (arrowhead 2A).

Figure 3. hydrolytic enzymes activities. A: cellulose (not cellulose)

Figure 5.:  Please give information about the medium.  Morphology of S. yangpuensis CM253. (A) colony morphology on ……: the single colony is nearly round, slightly convex, dry, with a white aerial hypha, and a pale-yellow basal mycelium.

References: Formal corrections are necessary.

Examples (corrections are indicated with red color):

1. Moratalla-López, N.; Bagur, M.J.; Lorenzo, C.; Martínez-Navarro, M.E.; Salinas, M.R.; Alonso, G.L. Bioactivity and bioavailability of the major metabolites of Crocus sativus L. flower. Molecules 2019, 24, doi:10.3390/molecules24152827.

2. Pitsikas, N. Crocus sativus L. extracts and its constituents crocins and safranal; potential candidates for schizophrenia  treatment? Molecules 2021, 26, doi:10.3390/molecules26051237.
